# ON THE RELATIONSHIP BETWEEN NEURAL MACHINE TRANSLATION AND WORD ALIGNMENT

## ABSTRACT

Prior researches suggest that attentional neural machine translation (NMT) is able to capture word alignment by attention, however, to our surprise, it almost fails for NMT models with multiple attentional layers except for those with a single layer. This paper introduce two methods to induce word alignment from general neural machine translation models. Experiments verify that both methods obtain much better word alignment than the method by attention. Furthermore, based on one of the proposed method, we design a criterion to divide target words into two categories (i.e. those mostly **c**ontributed **f**rom **s**ource "CFS" words and the other words mostly **c**ontributed **f**rom **t**arget "CFT" words), and analyze word alignment under these two categories in depth. We find that although NMT models are difficult to capture word alignment for CFT words but these words do not sacrifice translation quality significantly, which provides an explanation why NMT is more successful for translation yet worse for word alignment compared to statistical machine translation. We further demonstrate that word alignment errors for CFS words are responsible for translation errors in some extent by measuring the correlation between word alignment and translation for several NMT systems.

## 1 INTRODUCTION

Machine translation aims at modeling the semantic equivalence between a pair of source and target sentences (Koehn, 2009), and word alignment tries to model the semantic equivalence between a pair of source and target words (Och & Ney, 2003). As a sentence consists of words, word alignment is conceptually related to machine translation and such a relation can be traced back to the birth of statistical machine translation (SMT) (Brown et al., 1993), where word alignment is the basis of SMT models and its accuracy is generally helpful to improve translation quality (Koehn et al., 2003; Liu et al., 2005).

In neural machine translation (NMT), it is also important to study word alignment, because word alignment provides potential ways to understanding black-box NMT models and analyzing their translation errors (Ding et al., 2017). Unlike SMT, NMT models do not directly depend on word alignment and prior researches implicitly extract word alignment by attention (Bahdanau et al., 2014; Mi et al., 2016; Liu et al., 2016). Unfortunately, this method is not general for all NMT models. In particular, to our surprise, it can only obtain word alignment for NMT models with a single attentional layer, but fails for those with multiple attentional layers, which is the standard for state-of-the-art NMT systems such as TRANSFORMER (Vaswani et al., 2017).

In this paper, we propose two methods to induce word alignment from general NMT models and answer a fundamental question how much word alignment NMT models can learn. The first method explicitly builds a word alignment model between a pair of source and target word representations encoded by NMT models, and then it learns additional parameters for this word alignment model with the supervision from external aligners similar to (Mi et al., 2016) and (Liu et al., 2016). The second method is more intuitive and flexible: it is parameter-free and thus does not need retraining and external aligners. Its key idea is to measure the prediction difference of a target word if a source word is removed, inspired by (Arras et al., 2016; Zintgraf et al., 2017). Unlike the first method, the second one only depends on NMT models, and it thereby facilitates better understanding and interpreting NMT models. Experiments on an advanced NMT model show that both methods achieve much better word alignment than the method by attention.

However, word alignment obtained by our methods is still worse than statistical word aligners (Dyer et al., 2013; Och & Ney, 2003). This raises another natural question why advanced NMT models including less word alignment knowledge but deliver better translations than SMT models based on statistical aligners, which was observed in prior researches yet without deep interpretation (Tu et al., 2016; Liu et al., 2016). To answer this question, in the spirit of the proposed prediction difference (PD) method, we design a criterion to divide target words in the reference of a test sentence into two categories: those target words mostly contributed from source (CFS) words and the other target words mostly contributed from target (CFT) words in their histories. [1] Our experiments further demonstrate that NMT models capture good word alignment for the first category (CFS) words with accuracy comparable to a statistical aligner, while their word alignment for the second category (CFT) words is much worse. Thankfully, most target words in the second category (CFT) can be easily decoded by NMT models because of the strong language model effects implicitly learned by NMT models. Finally, we exploit the correlation between translation quality and word alignment quality for neural machine translation. We find that there is a weak correlation between translation and word alignment for all target words, yet a strong correlation between translation and word alignment for those target words mostly contributed from source words, after analyzing translations from several NMT systems. This finding suggests that word alignment errors for CFS words are responsible for translation errors in some extent.

This paper makes the following contributions:

- It proposes two better methods to acquire word alignment from general NMT models.

- It explains why NMT models deliver excellent translations no matter their worse word alignment compared to statistical machine translation.

- It empirically shows a strong correlation between translation quality and word alignment accuracy for those target words mostly contributed from source words.

## 2 PRELIMINARIES

### 2.1 NEURAL MACHINE TRANSLATION

Given a source sentence $\mathbf{x} = \langle \mathrm{x}_1, \cdots, \mathrm{x}_{|\mathbf{x}|} \rangle$ and a target sentence $\mathbf{y} = \langle \mathrm{y}_1, \cdots, \mathrm{y}_{|\mathbf{y}|} \rangle$, NMT aims at maximizing the following conditional probabilities: [2]

$$P\left(\mathbf{y} \mid \mathbf{x}\right) = \prod_{i=1}^{|\mathbf{y}|} P\left(\mathrm{y}_i \mid \mathbf{y}_{<i}, \mathbf{x}\right) = \prod_{i=1}^{|\mathbf{y}|} P\left(\mathrm{y}_i \mid \mathrm{s}_i^L\right), \tag{1}$$

where $\mathbf{y}_{<i} = \langle \mathrm{y}_1, \ldots, \mathrm{y}_{i-1} \rangle$ denotes a prefix of $\mathbf{y}$ with length $i-1$, and $\mathrm{s}_i^L$ is the decoding state of $\mathrm{y}_i$. Generally, the conditional distribution $P\left(\mathrm{y}_i \mid \mathrm{s}_i^L\right)$ is somehow modeled within an encoder-decoder framework. In encoding stage, the source sentence $\mathbf{x}$ are encoded as a sequence of hidden vectors $\mathbf{h}$ by an encoder according to specific NMT models, such as a multi-layer encoder consisting of recurrent neural network (RNN), convolutional neural network (CNN), or self-attention layer. In decoding stage, each decoding state $\mathrm{s}_i^L$ is computed by an $L$-layer decoder as follows:

$$\mathrm{s}_i^l = f\left(\mathrm{c}_i^l, \mathrm{s}_i^{l-1}, \mathbf{s}_{<i}^l, \mathbf{h}\right), \quad l \in \{1, \ldots, L\}, \quad \mathrm{s}_i^0 = \boldsymbol{y}_i, \tag{2}$$

where $\boldsymbol{y}_i$ is the word embedding of word $\mathrm{y}_i$, $f$ is a general function dependent on a specific NMT model, $\mathrm{c}_i^l$ is a context vector in $l^{\mathrm{th}}$ layer, computed from $\mathbf{h}$ and $\mathbf{s}_{<i}^l$ according to different NMT models. As the dominant models, attentional NMT models define the context vector $\mathrm{c}_i^l$ as a weighted sum of $\mathbf{h}$, where the weight $\boldsymbol{\alpha}_i^l = g\left(\mathrm{s}_i^{l-1}, \mathbf{s}_{<i}^l, \mathbf{h}\right)$ is defined by a similarity function $g$. Due to the space limitation, we refer readers to (Bahdanau et al., 2014), (Gehring et al., 2017) and (Vaswani et al., 2017) for the details on the definitions of $f$ and $g$.

---

[1] For a sentence $\langle \mathrm{y}_1, \mathrm{y}_2, \cdots, \mathrm{y}_i, \cdots, \mathrm{y}_{|\mathbf{y}|} \rangle$, a history word of $\mathrm{y}_i$ can be any $\mathrm{y}_k$ sufficing $k < i$.

[2] Throughout this paper, bold font such as $\mathbf{x}$ denote a sequence while regular font such as x denote an element which may be a scaler $x$, vector $\boldsymbol{x}$ or matrix $\boldsymbol{X}$.

## 2.2 ALIGNMENT BY ATTENTION

Since the attention weight $\boldsymbol{\alpha}_{i,j}^{l}$ measures the similarity between $s_i^{l-1}$ and $h_j$, it has been widely used to evaluate the word alignment between $y_i$ and $x_j$ (Bahdanau et al., 2014; Ghader & Monz, 2017). Once an attentional NMT model has been trained, one can easily extract word alignment $\boldsymbol{A}$ from the attention weight $\boldsymbol{\alpha}$ according to the style of maximum a posterior strategy (MAP) as follows:

$$\boldsymbol{A}_{i,j}(\boldsymbol{\alpha}) = \begin{cases} 1 & j = \arg\max_{j'} \boldsymbol{\alpha}_{i,j'} \\ 0 & o/w \end{cases}, \tag{3}$$

where $\boldsymbol{A}_{i,j} = 1$ indicates $y_i$ aligns to $x_j$. For NMT models with multiple attentional layers or multiple head attentional layers as in (Vaswani et al., 2017), we sum all attention weights with respect to all layers and heads to a single $\boldsymbol{\alpha}$ before MAP in equation 3.

## 3 METHODS TO INDUCING WORD ALIGNMENT

Although attention might obtain some word alignment as described in previous section, it is unknown whether NMT models contain more word alignment information than that obtained by attention. In addition, the method using attention is useful to induce word alignment for attentional NMT models, (more precisely, those models including a single attentional layer as shown in our experiments), whereas it is useless for general NMT models. In this section, in order to induce word alignment from general NMT models, we propose two different methods, which are agnostic to specific NMT models.

### 3.1 ALIGNMENT BY EXPLICIT ALIGNMENT MODEL

Given a source sentence $\mathbf{x}$, a target sentence $\mathbf{y}$, following (Liu et al., 2005) and(Taskar et al., 2005), we explicitly define a word alignment model as follows:

$$P\left(x_j \mid y_i, \mathbf{y}, \mathbf{x}; \boldsymbol{W}\right) = \frac{\exp\left(\delta\left(x_j, y_i, \mathbf{x}, \mathbf{y}; \boldsymbol{W}\right)\right)}{\sum_{j'=1}^{m} \exp\left(\delta\left(x_{j'}, y_i, \mathbf{x}, \mathbf{y}; \boldsymbol{W}\right)\right)}, \tag{4}$$

where $\delta\left(x_j, y_i, \mathbf{x}, \mathbf{y}; \boldsymbol{W}\right)$ is a distance function parametrized by $\boldsymbol{W}$. Ideally, $\delta$ is able to include arbitrary features such as IBM model 1 similar to (Liu et al., 2005). However, as our goal is not to achieve the best word alignment but to focus on that captured by an NMT model, we only consider these features completely learned in NMT. Hence, we define the

$$\delta\left(x_j, y_i, \mathbf{x}, \mathbf{y}; \boldsymbol{W}\right) = \left(\boldsymbol{x}_j \| \boldsymbol{h}_j\right)^{\top} \boldsymbol{W} \left(\boldsymbol{y}_i \| \boldsymbol{s}_i^{L}\right), \tag{5}$$

where $\boldsymbol{x}_j$ and $\boldsymbol{y}_i$ are word embeddings of $x_j$ and $y_i$ learned in NMT, $\boldsymbol{h}_j$ is the hidden unit of $x_j$ in the encoding network and $\boldsymbol{s}_i^{L}$ is the hidden unit of $y_j$ in the decoding network, $\|$ denotes the concatenation of a pair of column vectors of dimension $d$, and $\boldsymbol{W}$ is a matrix of dimension $2d \times 2d$.

The explicit word alignment model is trained by maximizing the objective function with respect to $\boldsymbol{W}$:

$$\max_{\boldsymbol{W}} \sum_{\mathbf{x},\mathbf{y}} \sum_{\forall j,i: \boldsymbol{A}_{ij}^{\mathrm{ref}}=1} \log P\left(x_j \mid y_i, \mathbf{y}, \mathbf{x}; \boldsymbol{W}\right), \tag{6}$$

where $\boldsymbol{A}_{ij}^{\mathrm{ref}}$ is the reference alignment between $x_j$ and $y_i$ for a sentence pair $\mathbf{x}$ and $\mathbf{y}$. As the number of elements in $\boldsymbol{W}$ is up to one million (i.e., $(2 \times 512)^2$), it is not feasible to train it using a small dataset with gold alignment. Therefore, following (Mi et al., 2016; Liu et al., 2016), we run statistical word aligner such as FAST ALIGN (Dyer et al., 2013) on a large corpus and then employ resulting word alignment as the silver alignment $\boldsymbol{A}^{\mathrm{ref}}$ for training. Note that our goal is to quantify word alignment learned by an NMT model, and thus we only treat $\boldsymbol{W}$ as the parameter to be learned, which differs from the joint training all parameters including those from NMT models as in (Mi et al., 2016; Liu et al., 2016).

After training, one obtains the optimized $\boldsymbol{W}$ and then easily infers word alignment for a test sentence pair $\langle \mathbf{x}, \mathbf{y} \rangle$ via the MAP strategy as defined in equation 3 by setting $\boldsymbol{\alpha}_{i,j'} = P\left(x_{j'} \mid y_i, \mathbf{y}, \mathbf{x}; \boldsymbol{W}\right)$.

Note that if word embeddings and hidden units learned by NMT models capture enough information for word alignment, the above method can obtain excellent word alignment. However, because the dataset for supervision in training definitely include some data intrinsic word alignment information, it is unclear how much word alignment is only from NMT models. Therefore, we propose the other method which is parameter-free and only dependent on NMT models themselves.

## 3.2 Alignment by Prediction Difference

The intuition to this method is that if $y_i$ aligns to $x_j$, the relevance between $y_i$ and $x_j$ should be much higher than that between $y_i$ and any other $x_k$ with $k \neq j$. Therefore, the key to our method is that how to measure the relevance between $y_i$ and $x_j$.

**Sampling method** Zintgraf et al. (2017) propose a principled method to measure the relevance between a pair of tokens in input and output. It is estimated by measuring how the prediction of $y_i$ in the output changes if the input token $x_j$ is unknown. Formally, the relevance between $y_i$ and $x_j$ for a given sentence pair $\langle \mathbf{x}, \mathbf{y} \rangle$ is defined as follows:

$$R\left(y_i, x_j, \mathbf{x}, \mathbf{y}\right) = P\left(y_i \mid \mathbf{y}_{<i}, \mathbf{x}\right) - P\left(y_i \mid \mathbf{y}_{<i}, \mathbf{x}_{\setminus j}\right), \tag{7}$$

with

$$P\left(y_i \mid \mathbf{y}_{<i}, \mathbf{x}_{\setminus j}\right) = \sum_{\mathbf{x}} P\left(\mathbf{x} \mid \mathbf{y}_{<i}, \mathbf{x}_{(j,\varnothing)}\right) P\left(y_i \mid \mathbf{y}_{<i}, \mathbf{x}_{(j,\mathbf{x})}\right)$$

$$\approx \mathbb{E}_{\mathbf{x} \sim P(\mathbf{x})}\left[P\left(y_i \mid \mathbf{y}_{<i}, \mathbf{x}_{(j,\mathbf{x})}\right)\right], \tag{8}$$

where $\mathbf{x}_{(j,\mathbf{x})} = \langle x_1, \ldots, x_{j-1}, x, x_{j+1}, \ldots, x_{|\mathbf{x}|} \rangle$ denotes the sequence by replacing $x_j$ with $x$ in $\mathbf{x}$, particularly $\mathbf{x}_{(j,\varnothing)} = \langle x_1, \ldots, x_{j-1}, x_{j+1}, \ldots, x_{|\mathbf{x}|} \rangle$ denotes the sequence by removing $x_j$ from $\mathbf{x}$, $P(y_i \mid \mathbf{y}_{<i}, \mathbf{x})$ is defined in equation 1 and $P\left(x \mid \mathbf{y}_{<i}, \mathbf{x}_{(j,\varnothing)}\right)$ is approximated by the empirical distribution $P(\mathbf{x})$, which can be considered as the 1-gram language model for the source side of the training corpus. Unlike a computer vision task in (Zintgraf et al., 2017), the size of source vocabulary in NMT is up to 30000 and thus summation over this large vocabulary is challenging in computational efficiency. As a result, we only sample multiple words to approximate the expectation in equation 8 by Monte Carlo (MC) approach.

**Deterministic method** Inspired by the idea of dropout (Srivastava et al., 2014), we measure the relevance by disabling the connection between $x_j$ and the encoder network in a deterministic way. Formally, $R\left(y_i, x_j, \mathbf{x}, \mathbf{y}\right)$ is directly defined via dropout effect on $x_j$ as follows:

$$R\left(y_i, x_j, \mathbf{x}, \mathbf{y}\right) = P\left(y_i \mid \mathbf{y}_{<i}, \mathbf{x}\right) - P\left(y_i \mid \mathbf{y}_{<i}, \mathbf{x}_{(j,\mathbf{0})}\right), \tag{9}$$

where $\mathbf{x}_{(j,\mathbf{0})}$ denotes the sequence by replacing $x_j$ with a word whose embedding is a zero vector. In this way, the computation in equation 9 is much faster than the Monte Carlo sampling approach involving multiple samples. It is worth mentioning that equation 9 resembles the Monte Carlo sampling approach with a single sample in calculation, but it is significantly better than MC with a single sample in alignment quality and is very close to MC approach with enough samples, as to be shown in our experiments.

Note that $R(y_i, x_j, \mathbf{x}, \mathbf{y}) \in [-1, 1]$, where $R(y_i, x_j, \mathbf{x}, \mathbf{y}) = 1$ means $i^{\text{th}}$ target word is totally determined by the $j^{\text{th}}$ source word; $R(y_i, x_j, \mathbf{x}, \mathbf{y}) = -1$ means $i^{\text{th}}$ target word and $j^{\text{th}}$ source word are mutual exclusive; $R(y_i, x_j, \mathbf{x}, \mathbf{y}) = 0$ means $j^{\text{th}}$ source word do not affect generating $i^{\text{th}}$ target word. In order to obtain word alignment, after collecting $R(y_i, x_j, \mathbf{x}, \mathbf{y})$ for $x_j$, $y_i$, $\mathbf{x}$ and $\mathbf{y}$, one can easily infer word alignment via the MAP strategy as defined in equation 3 by setting $\boldsymbol{\alpha}_{i,j'} = R(y_i, x_{j'}, \mathbf{x}, \mathbf{y})$.

**Remark** The above $R(y_i, x_j, \mathbf{x}, \mathbf{y})$ in equation 7 quantifies the relevance between a target word $y_i$ and a source word $x_j$. Similarly, one can quantify the relevance between $y_i$ and its history word $y_k$ as follows:

$$R_o\left(y_i, y_k, \mathbf{x}, \mathbf{y}\right) = P\left(y_i \mid \mathbf{y}_{<i}, \mathbf{x}\right) - P\left(y_i \mid \mathbf{y}_{<i(k,\mathbf{0})}, \mathbf{x}\right), \tag{10}$$

where $R_o$ indicates the relevance between two target words $y_i$ and $y_k$ with $k < i$, and $P(y_i \mid \mathbf{y}_{<i(k,\mathbf{0})}, \mathbf{x})$ is obtained by disabling the connection between $y_k$ and the decoder network, similar to

$P\left(\mathbf{y}_i \mid \mathbf{y}_{<i}, \mathbf{x}_{(j,\mathbf{0})}\right)$. Unlike $R(\mathbf{y}_i, \mathbf{x}_j, \mathbf{x}, \mathbf{y})$ capturing word alignment information, $R_o(\mathbf{y}_i, \mathbf{y}_k, \mathbf{x}, \mathbf{y})$ is able to capture word allocation in a target sentence and it will be used to answer a fundamental question why NMT models yields better translation yet worse word alignment compared with SMT in section of experiments.

## 4 Experiments

In this section, we will empirically explore the following questions through experiments:

1. Does attention really capture word alignment for attentional NMT models?
2. How much word alignment do NMT models learn?
3. Why NMT models deliver better translation yet worse word alignment than SMT?
4. What is the relationship between translation quality and word alignment quality?

To answer these questions, we conduct experiments on Chinese-to-English dataset, which includes many reorderings and thereby is challenging for word alignment and translation tasks. Translation quality is evaluated with case-insensitive 4-gram BLEU (Papineni et al., 2002), implemented by *multi-bleu.perl*, and alignment performance is evaluated by AER (Mihalcea & Pedersen, 2003; Koehn, 2009). In the following experiments and analyzes, both BLEU and AER are shown in percentage.

### 4.1 Setting

The training data consists of 1.8M sentence pairs from Chinese-to-English task of NIST2008 Open Machine Translation Campaign with 40.1M Chinese words and 48.3M English words respectively. The development set is chosen as NIST2002, and the test set is NIST2005. To make NMT model capable of open-vocabulary translation, all the datasets are pre-processed by Byte Pair Encoding (BPE) (Sennrich et al., 2015) with 32K merging operations. [3] To measure word alignment quality, we report AER on NIST05 test set, whose reference alignment was manually annotated by experts (Liu et al., 2016). The experiments are based on an alignment baseline and three strong translation baselines from NMT and SMT:

- FAST ALIGN (Dyer et al., 2013): a simple, fast, unsupervised word aligner.
- MOSES (Koehn et al., 2007): an open source phrase based translation system with default configuration, whose translation tables are derived from FAST ALIGN.
- NEMATUS (Sennrich et al., 2017): an open source neural machine translation implementation of RNN based sequence-to-sequence model.
- TRANSFORMER (Vaswani et al., 2017): a novel neural network architecture for language understanding implemented by Zhang et al. (2017).

More details on training these systems are described in Appendix A.

### 4.2 Does Attention Really Capture Alignment?

To our surprise, word alignment by attention is with AER around 83 for TRANSFORMER with six attentional layers: we examined averaged alignment and that from each attentional layer, but any of their alignment is worse.

Since the bilingual corpus intrinsically includes word alignment in some extent, word alignment by attention should be better than the data intrinsic alignment if attention indeed captures alignment. To obtain the data intrinsic word alignment, we calculate point-wise mutual information (PMI) from the bilingual corpus and then infer word alignment for each bilingual sentence by using

---

[3]Throughout this paper, both BLEU and AER are evaluated after restoring BPE. Particularly for restoring BPE before acquiring alignment via MAP strategy in equation 3, $\boldsymbol{\alpha}$ is firstly merged along target tokens by averaging then merged along source tokens by summation.

Table 1: Word alignment captured by attention in TRANSFORMER and NEMATUS

| Methods | AER | BLEU |
|---|---|---|
| PMI | 65.66 | None |
| TRANSFORMER-L6 (attention) | 83.07[*] | 46.95 |
| TRANSFORMER-L1 (attention) | 56.49 | 36.51 |
| NEMATUS (attention) | 50.60 | 36.82 |

[*] Attention captures less word alignment than PMI for NMT models with multiple attentional layers.

the MAP strategy as in equation 3. [4] Referring to Table 1, alignment captured by attention of six-layer TRANSFORMER (TRANSFORMER-L6) is obviously worse than alignment by PMI. Hence, it is hard to conclude attention from the standard six-layer TRANSFORMER captures word alignment.

As shown in Table 1, NEMATUS using single attentional layer of RNN is able to capture better word alignment than PMI, so the architecture of multiple attentional layers in TRANSFORMER may be the reason why its attention fails to capture word alignment. Therefore, we reset TRANSFORMER to include one attentional layer in both encoder and decoder and retrain it on the same dataset. We find that the TRANSFORMER with a single attentional layer (TRANSFORMER-L1) can produce much more reasonable alignment than TRANSFORMER-L6, even if its translation quality substantially decreases. In this sense, we believe that the hidden units $h_j$ and $s_i^l$ in equation 2 represent the information of the entire sequence $\mathbf{x}$ and $\mathbf{y}$ instead of the information particularly emphasizing the words $x_j$ and $y_i$, and thereby attention between $h_j$ and $s_i^l$ is not necessary to indicate the word alignment between $x_j$ and $y_i$ especially for multi-layer attention models.

### 4.3 HOW MUCH ALIGNMENT CAN NMT LEARN?

Previous subsection shows alignment by attention may not be a well-defined method to acquiring alignment particularly for multi-layers models, and thus we need better methods to tell how much alignment NMT models can learn. In this subsection, we analyze performances of the two proposed methods, which includes an explicit alignment model and prediction difference based on TRANSFORMER-L6.

Table 2: AER of the proposed methods on TRANSFORMER-L6

| Methods | AER |
|---|---|
| FAST ALIGN | 36.57 |
| NEMATUS (attention) | 50.60 |
| TRANSFORMER-L1 (attention) | 56.49 |
| PMI (FAST ALIGN) | 60.18 |
| Explicit Alignment Model | 38.88 |
| Prediction Difference | 41.77 |

**Explicit Alignment Model** As the explicit model employs the silver alignment dataset from FAST ALIGN for training its parameters, its final AER includes contributions from both the aligned data and the model. We analyze the alignment performance from the model itself by comparing with PMI (FAST ALIGN), which is similar to PMI but calculates co-occurrence of a word pair aligned by FAST ALIGN in a bilingual sentence. [4] As shown in Table 2, the explicit model outperforms PMI (FAST ALIGN), which indicates that the six-layer TRANSFORMER indeed learns word alignment information. In addition, AER by the explicit model is better than attention method over both one layer TRANSFORMER and RNN models.

One might argue that the reason why the explicit model yields excellent alignment is attributed to the supervision from silver alignment dataset. However, with the same amount of supervision, TRANSFORMER-L1 generates much worse alignment than TRANSFORMER-L6 and it is only comparable to PMI (FAST ALIGN) in AER, as shown in Table 3. This suggests that supervision is

---

[4]More details in Appendix C.

Table 3: Explicit alignment model on different translation models

| Models | | TRANSFORMER | | | | | | PMI |
|---|---|---|---|---|---|---|---|---|
| | | L1 | L2 | L3 | L4 | L5 | L6 | (FAST ALIGN) |
| AER | EAM[*] | 54.50 | 47.94 | 40.47 | 38.40 | 38.80 | 38.88 | 60.18 |
| | Attention | 56.49 | 76.96 | 81.23 | 81.83 | 87.15 | 86.87 | |
| BLEU | | 36.51 | 44.83 | 45.63 | 47.19 | 46.35 | 46.95 | N/A |

[*] **EAM** is the abbreviation of "Explicit Alignment Model".

not enough to obtain good alignment and the hidden units learned by a translation model indeed implicitly capture alignment knowledge.

**Prediction Difference**    As shown in Table 2, prediction difference delivers better word alignment than the data intrinsic word alignment, i.e. PMI (FAST ALIGN), and this result gives a strong indicator that TRANSFORMER with multiple attentional layers indeed induces reasonable alignment even though the alignment is not captured by its attention. In addition, prediction difference is able to capture better word alignment than attention in the state of the art RNN model NEMATUS. However, both explicit model and prediction difference are worse than statistical word aligner FAST ALIGN in terms of AER.

Table 4: Comparison between sampling and deterministic methods for prediction difference

| Methods | Sampling method | | | | | Deterministic method |
|---|---|---|---|---|---|---|
| Sample size | 1 | 2 | 4 | 20 | 50 | |
| AER | 44.92 | 43.30 | 42.42 | 41.83 | 41.73 | 41.77 |
| Variance | 0.004 | $< 10^{-5}$ | $< 10^{-5}$ | $< 10^{-5}$ | $< 10^{-5}$ | N/A |

[*] Results are measured on TRANSFORMER-L6.

Prediction different can be implemented by sampling method or deterministic method. As shown in Table 4, the alignment performance of sampling method is improving as growing of the sample size, because the accuracy of Monte Carlo approach is dependent on the number of samples. And no matter what sample size is, the variance of AER is always ignorable. The reason might be $\arg\max$ operation in equation 3 eliminate the fluctuation of probability matrix. Although using large sample size can achieve nice alignment performance, but it will cost too much computation. Fortunately, the deterministic method can also achieve the nice alignment performance with the same computational cost as sample size equals one in sampling method. As a result, we employ deterministic version as the default for prediction difference in this paper.

Although explicit model achieves better word alignment result than prediction difference, it is difficult to interpret and understand neural machine translation through word alignment from explicit model. The main reason is that explicit model relies on an external aligned dataset with guidance from statistical word aligner FAST ALIGN, and thus the characteristic of its alignment result are similar to that of FAST ALIGN, leading to interpretation biased to FAST ALIGN. On the other hand, prediction difference only relies on prediction from a neural model to define the relevance, it has been successfully used to understand and interpret a neural model (Zintgraf et al., 2017). Therefore, in the rest of this section, we try to understand TRANSFORMER by using prediction difference in the rest of this section.

### 4.4    DOES ALIGNMENT ERROR AFFECT TRANSLATION?

It is still unreasonable that TRANSFORMER is good at translation even though messes up with word alignment. In order to reveal this observation, we firstly divide the target words into two categories:

- Contributing from source (CFS): the prediction of a target word can be mostly attributed to the appearance of a source side word.

- Contributing from target (CFT): the prediction of a target word can be mostly attributed to the appearance of a target side word.

Specifically, we employ prediction difference to define CFS and CFT as follows:

$$\begin{aligned} \text{CFS:} \quad & \left\{ \mathbf{y}_i \in \mathbf{y} \mid \max_{\mathbf{x} \in \mathbf{x}} R(\mathbf{y}_i, \mathbf{x}, \mathbf{x}, \mathbf{y}) - \max_{\mathbf{y} \in \mathbf{y}_{<i}} R_o(\mathbf{y}_i, \mathbf{y}, \mathbf{x}, \mathbf{y}) > \epsilon \right\}, \\ \text{CFT:} \quad & \left\{ \mathbf{y}_i \in \mathbf{y} \mid \max_{\mathbf{y} \in \mathbf{y}_{<i}} R_o(\mathbf{y}_i, \mathbf{y}, \mathbf{x}, \mathbf{y}) - \max_{\mathbf{x} \in \mathbf{x}} R(\mathbf{y}_i, \mathbf{x}, \mathbf{x}, \mathbf{y}) > \epsilon \right\}. \end{aligned} \tag{11}$$

where $\epsilon \in [0, 1)$ is a probability margin between CFS and CFT words.

Table 5: Word alignment quality affects translation quality

| Methods | Target Words[*] | AER | Translation Recall[†] |
|---|---|---|---|
| PD & TRANSFORMER-L6 | Overall | 41.77 | 63.81 |
| | CFS | 33.95 | 64.51 |
| | CFT | 63.28 | 62.10 |
| FAST ALIGN & MOSES | Overall | 36.57 | 60.76 |
| | CFS | 31.66 | 61.74 |
| | CFT | 50.80 | 58.42 |

[*] Overall target words, 70.71% target words belong to CFS and 29.29% target words belong to CFT.
[†] Translation recall measures the percentages of target words in reference recalled by decoding.

After dividing the target words into two categories of CFS words and CFT words according to the criterion defined above, [5] we calculate their percentages and find that the ratio between CFS and CFT is about 7:3. This result suggests that TRANSFORMER employs more information from source side than target side for translation and it is more important for TRANSFORMER to make better use of source side information.

In addition, it is shown in Table 5 that AER of CFS words by prediction difference improves to 33.95 comparable to AER of CFS by FAST ALIGN, even though AER by prediction difference on overall targets is substantially worse than AER by FAST ALIGN. We further compare word alignment from prediction difference and that from FAST ALIGN according to the CFS words or the CFT words and the results are shown in Table 5. We find that prediction difference captures good alignment for CFS words as FAST ALIGN does, but its alignment for CFT words is much worse than FAST ALIGN. Additionally, we evaluate the effects of translation quality on both the CFS words and the CFT words by translation recall, which measures the percentages of target words in references recalled by TRANSFORMER and MOSES after decoding. As shown in Table 5, although TRANSFORMER induces worse alignment for the CFT words, it successfully decodes 62% of the CFT words, which is comparable to the percentages of the CFS words. These results demonstrate that TRANSFORMER is able to easily translate the CFT words by using target history words thanks to its strong language model effect, even if it involves noise source information due to inaccurate alignment. Therefore, CFT words might be the reason why TRANSFORMER yields better translation yet worse word alignment compared to SMT models.

### 4.5 CORRELATION BETWEEN TRANSLATION AND ALIGNMENT

We train six translation systems on the same dataset: they share the same model architecture but use encoder-decoder layer number ranging from 1 to 6. As there is only a single manually aligned dataset (i.e. NIST2005), the result might be highly dependent on the specific dataset. Inspired by (Koehn, 2004), we randomly sample 1200 datasets with replacement from NIST2005, each of which includes the same number of the sentences as NIST2005. We report their BLEU points and AER captured by prediction difference on the 1200 sampled datasets. Figure 1(a) shows the correlation between translation quality and alignment error rate is only about $-0.45$. This result indicates a weak correlation between translation and alignment.

Because CFT words yield worse alignment but are easy to translate as observed before, we propose to modify AER on CFS words only. However, since CFS words defined in previous section are dependent on a specific translation model whereas we employ six translation models, there will be a bias for the correlation between translation and alignment on CFS words for different translation models. Hence, we redefine CFS words using two PMI statistics based on a pair of source and target words and a pair of a target word and its history target word, [4] by replacing $R$ and $R_o$ with these

---

[5]Without affecting main conclusions, $\epsilon = 0$ in this experiment for covering all words in analysis. Experiments with different margins are in Appendix D.

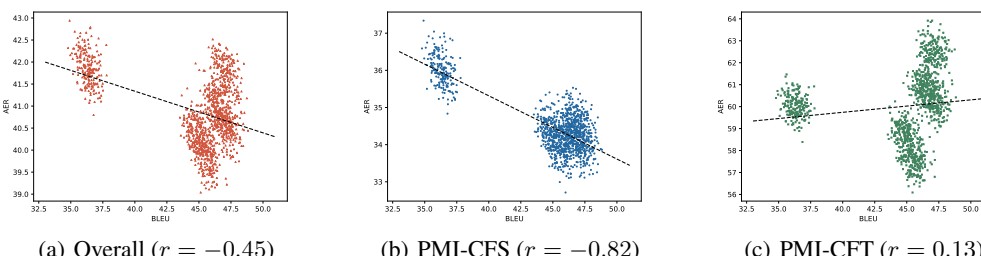

(a) Overall ($r = -0.45$)      (b) PMI-CFS ($r = -0.82$)      (c) PMI-CFT ($r = 0.13$)

Figure 1: Pearson correlation $r$ between translation quality and word alignment captured by the prediction difference method for Overall, CFS and CFT target words.

two PMI statistics in equation 11. It is known that the redefined CFS and CFT are dependent on the training data yet independent on any specific model. In order to avoid redundant concepts, they are still mentioned as CFS and CFT words as before.

In Figure 1(b), one can see that the modified AER on CFS words correlates well with BLEU points, and such a correlation is up to $-0.82$. Additionally, as shown in Figure 1(c), the correlation between translation and alignment for CFT words is around zero and even positive. This indicates that correct alignment of CFT words is not necessary to lead to improved translation quality, because these CFT words are easily figured out by the strong language model effects encoded in NMT models. Together with the result shown in Table 5, improving word alignment for CFS words is potential to improve translation quality for neural machine translation.

## 5 RELATED WORK

In NMT, there are many notable researches which mention word alignment captured by attention in some extent. For example, (Bahdanau et al., 2014) is the first work to show word alignment examples by using attention in an NMT model. Tu et al. (2016) quantitatively evaluate word alignment captured by attention and find that its quality is much worse than statistical word aligners. Motived by this finding, Chen et al. (2016), Mi et al. (2016) and Liu et al. (2016) improve attention with the supervision from silver alignment results obtained by statistical aligners, in the hope that the improved attention leads to better word alignment and translation quality consequently. Despite the close relation between word alignment and attention, Koehn & Knowles (2017) and Ghader & Monz (2017) discuss the differences between word alignment and attention in NMT. All these works study word alignment for the same kind of NMT models, i.e. that with a single attentional layer. One of our contribution is that we propose model-agnostic methods to study word alignment in a general way which deliver better word alignment quality than attention method. Moreover, for the first time, we further explain and quantify the relationship between translation quality and word alignment for an NMT model.

The prediction difference method in this paper actually provides an avenue to understand and interpret neural machine translation models. Therefore, it is closely related to many works on visualizing and interpreting neural networks (Lei et al., 2016; Bach et al., 2015; Zintgraf et al., 2017). Indeed, our method is inherited from (Zintgraf et al., 2017), and our advantage is that it is computationally efficient particularly for those tasks with a large vocabulary. In sequence-to-sequence tasks, Ding et al. (2017) focus on model interpretability by modeling how influence propagates across hidden units in networks, which is often too restrictive and challenging to achieve as argued by (Alvarez-Melis & Jaakkola, 2017). Instead, Alvarez-Melis & Jaakkola (2017) concentrate on prediction interpretability with only oracle access to the model generating the prediction. To achieve this effect, they propose a casual learning framework to measure the relevance between a pair of source and target words. Our method belongs to the type of prediction interpretability similar to (Alvarez-Melis & Jaakkola, 2017), but ours is a unified and parameter-free method rather than a pipeline and parameter-dependent one. In addition, both Ding et al. (2017) and Alvarez-Melis & Jaakkola (2017) qualitatively demonstrate interpretability by showing some sentences, while we exhibit the interpretability by quantitatively analyzing all sentences in a test set.

## 6 CONCLUSION AND FUTURE WORK

This paper points out that attention is insufficient to induce word alignment or even surprisingly fails for general NMT models. Therefore, it proposes two better methods to induce word alignment than the method by attention for general NMT models. Based on one of the proposed method, it divides target words into two categories including CFS and CFT, and shows that NMT models yield excellent word alignment on CFS words but worse word alignment on CFT words, which do not significantly sacrifice translation quality. This explains a fundamental question why NMT delivers better translation yet worse word alignment than its SMT counterpart. Finally, this paper empirically demonstrates that word alignment errors for CFS words are responsible for translation errors in some extent by measuring the correlation between word alignment quality and translation quality. In the future, we believe that more work is interesting to analyze translation errors such as those errors for CFT words. In addition, we will investigate solutions to improving NMT models, in the hope of using source context and target history context in a more robust manner for better predicting CFS and CFT words.

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

## A  TRAINING DETAILS

We implemented the proposed methods to induce word alignment on TRANSFORMER, since it is the most popular NMT model nowadays. For training MOSES, we use all 1.8M sentences from the corpus, and we train a 4-gram language model based on the target side of its training data. For training both NMT models, only the sentences of length up to 256 tokens are used, with no more than $2^{15}$ tokens in a batch. The dimension of both word embeddings and hidden states are 512. Both encoder and decoder have 6 layers by default, and adopt multi-head attention with 8 heads. The beam size for decoding is 4, and the loss function is optimized by Adam (Kingma & Ba, 2014), where $\beta_1 = 0.9$, $\beta_2 = 0.98$ and $\epsilon = 10^{-9}$. Particularly for the explicit alignment model, the alignment reference is produced by FAST ALIGN.

Note that MOSES and NEMATUS achieve BLEU points on NIST2005 test set comparable to those reported in recent works using the similar corpora (Tu et al., 2016; Liu et al., 2016; Zhou et al., 2017), and TRANSFORMER achieves much higher performance (i.e. 47 BLEU points). This shows that our MT models are well-trained.

## B  CASE STUDY

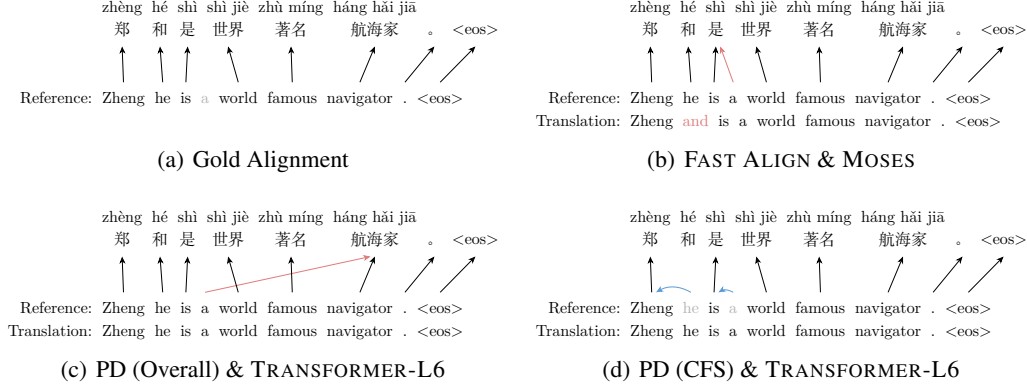

(a) Gold Alignment

(b) FAST ALIGN & MOSES

(c) PD (Overall) & TRANSFORMER-L6

(d) PD (CFS) & TRANSFORMER-L6

Figure 2: An example of word alignment and translation produced by SMT and NMT systems. Red arrow means wrong alignment and blue arrow means the prediction is attributed to a target word. The word in light font do not align to any source word, while red word means wrong translation.

Here is an example to demonstrate the results of previous experiments intuitively. As shown in Figure 2(a), the word 'a' should not be aligned to any source word. However, in Figure 2(b) FAST ALIGN wrongly aligned 'a' to 'shì', and in Figure 2(c) PD makes the same mistake. Fortunately, as shown in Figure 2(d), if we only consider alignment of words in CFS, 'a' is superbly not aligned to any source word because it belongs to CFT. In terms of translation, TRANSFORMER translates perfectly, yet MOSES wrongly translate 'hé' into 'and' due to 'hé' mostly means 'and' ignoring the context in Chinese. Instead of depending on phrase table, as shown in Figure 2(d), TRANSFORMER successfully translate 'hé' into the given name 'he' referring to the surname 'Zheng' in translation history, thanks to its more powerful language model.

## C  POINTWISE MUTUAL INFORMATION AMONG WORDS

Pointwise mutual information (PMI) measures the relevance of two discrete random variables, which is defined as

$$\mathrm{PMI}(\mu, \nu) = \log \frac{P(\mu, \nu)}{P(\mu) \cdot P(\nu)} = \log Z + \log \frac{C(\mu, \nu)}{C(\mu) \cdot C(\nu)}, \tag{12}$$

where $C(\mu, \nu)$ is a function for counting occurrence of the pair $(\mu, \nu)$ according to different scenarios, and $Z$ is the normalizer, i.e. the total number of all possible $(\mu, \nu)$ pairs. In this paper, we define three types of PMI according to different definitions of $C(\mu, \nu)$ in the three scenarios as follows.

**PMI on Bilingual Data** In this scenario, a set of bilingual sentences is given. For a given bilingual sentence $\langle \mathbf{x}, \mathbf{y} \rangle$, $C(\mathrm{y}_i, \mathrm{x}_j)$ is added by one if both $\mathrm{y}_i \in \mathbf{y}$ and $\mathrm{x}_j \in \mathbf{x}$.

**PMI on Word Aligned Bilingual Data** In this scenario, a set of word aligned bilingual sentences is given. That is, for a bilingual sentence, words in its target side may align to words in its source side. For a given word aligned bilingual sentence $(\mathbf{x}, \mathbf{y})$, $C(\mathrm{y}_i, \mathrm{x}_j)$ is added by one if $\mathrm{y}_i \in \mathbf{y}$ and $\mathrm{x}_j \in \mathbf{x}$ and $\mathrm{y}_i$ aligns to $\mathrm{x}_j$.

**PMI between a Word and Its History Word on Monolingual Data** In this scenario, a set of monolingual sentences is given. For a given monolingual sentence $\mathbf{y}$, $C(\mathrm{y}_k, \mathrm{y}_i)$ is added by one if $\mathrm{y}_k \in \mathbf{y}$ and $\mathrm{y}_i \in \mathbf{y}$ with $k < i$.

# D  DIFFERENT MARGINS FOR DIVIDING CFS AND CFT

In equation 11, different margins will partition CFS and CFT differently. As growing of the margin $\epsilon$, the partition of CFS and CFT becomes more confident. As shown in Table 6, more confident CFS words can achieve better alignment performance and translation performance. But the translation recall is still similar between CFS and CFT words, despite the big gap of AER between CFS and CFT words.

Table 6: Word alignment and translation quality under different partitions of CFS and CFT

| $\epsilon$ | Target Words | AER | Translation Recall | Proportion |
|---|---|---|---|---|
| $10^{-4}$ | CFS | 31.64 | 65.54 | 65.61% |
| | CFT | 62.91 | 66.39 | 24.66% |
| $10^{-3}$ | CFS | 30.33 | 67.82 | 60.40% |
| | CFT | 63.29 | 69.40 | 22.04% |
| $10^{-2}$ | CFS | 28.26 | 71.56 | 51.53% |
| | CFT | 64.22 | 73.76 | 17.56% |
| $10^{-1}$ | CFS | 22.87 | 78.59 | 34.85% |
| | CFT | 64.13 | 78.33 | 10.39% |

\* Results are measured on TRANSFORMER-L6.

