# OpenReview forum: "On the Relationship between Neural Machine Translation and Word Alignment"
_ICLR.cc/2019/Conference_

### Official Review · AnonReviewer1 · 2018-11-01
**Interesting interpretability work cast as bilingual word alignment.**

**Rating:** 6
**Confidence:** 4

**Review:**

This paper sets out to build good bilingual word alignments from the information in an NMT system (both Transformer and RNN), where the goal is to match human-generated word-alignments as measured by AER. At least that’s how it starts. They contribute two aligners: one supervised aligner that uses NMT source and target representations as features and is trained on silver data generated by FastAlign, and one interpretability-based aligner that scores the affinity of a source-target word-pair by deleting the source word (replacing its embedding with a 0-vector) and measuring the impact on the probability of the target word. These are both shown to outperform directly extracting alignments from attention matrices by large margins. Despite the supervised aligner getting better AER, the authors proceed to quickly discard it as they dive deep on the interpretability approach, applying it also to target-target word pairs, and drawing somewhat interesting conclusions about two classes of target words: those that depend most of source context and those that depend most on target context.

Ultimately, this paper’s main contribution is its subtraction-based method for doing model interpretation. Its secondary contributions are the idea of evaluating this interpretation method empirically using human-aligned sentence pairs, and the idea of using the subtraction method on target-target pairs. The conclusion does a good job of emphasizing these contributions, but the abstract and front-matter do not. Much of the rest of the paper feels like a distraction. Overall, I believe the contributions listed above are valuable, novel and worth publishing. I can imagine using this paper’s techniques and ideas in my own research.

Specific concerns:

The front-matter mentions ‘multiple attention layers’. It would probably be a good idea to define this term carefully, as there are lots of things that could fit: multiple decoder layers with distinct attentions, multi-headed attention, etc.

In contrast to what is said in the introduction, GNMT as described in the Wu et al. 2016 paper only calculates attention once, based on the top encoder layer and the bottom decoder layer, so it doesn’t fit any definition of multiple attention layers.

Equation (1) and the following text use the variable L without defining it.

‘dominative’ -> ‘dominant’

Is there any way to generate a null alignment with Equation 3? That is, a target word that has no aligned source words? If not, that is a major advantage for FastAlign.

Similarly, what exactly are you evaluating when you evaluate FastAlign? Are you doing the standard tricks from the phrase-based days, and generating source->target and target->source models, and combining their alignments with grow-diag-final? If so, you could apply the same tricks to the NMT system to help even the playing field. Maybe this isn’t that important since the paper didn’t win up being about how to build the best possible word aligner from NMT (which I think is for the best).

I found Equations (7) and (8) to be confusing and distracting. I understand that you were inspired by Zintgraf’s method, but the subtraction-based method you landed on doesn’t seem to have much to do with the original Zintgraf et al. approach (and your method is much easier to the understand in the context of NMT than theirs). Likewise, I do not understand why you state, “we take the uniform distribution as P(x) regarding equation 8 for simplicity” - equation 9 completely redefines the LHS of equation 8, with no sum over x and no uniform distribution in sight.

The Data section of 4.1 never describes the NIST 2005 hand-aligned dataset.

The conclusions drawn at the end of 4.4 based on ‘translation recall’ are too strong. What we see is that the Transformer outperforms Moses by 2.8 onCFS, and by 3.7 on CFT. This hardly seems to support a claim that CFT words are the reason why Transformer yields better translation.

4.5 paragraph 1: there is no way to sample 12000 datasets without replacement from NIST 2005 and have the samples be the same size as NIST 2005. You must mean “with replacement”?

---

> ### Author Response · Authors · 2018-11-15
> **New manuscript with more clear descriptions has been uploaded**
>
> Thanks for your review and constructive comments. We address your concerns
> below.
>
> You mentioned "multiple attention layers", which might be the reason of
> terrible AER of attention-based alignment. As shown in Table 1, an acceptable
> alignment can be induced form a single layer multi-head attention, but cannot
> from a multi-layer multi-head attention. Therefore, we ascribe the degradation
> of alignment quality to "multiple attention layers" rather than "the forms of
> attention". The reason behind is multi-layer attention disrupt the order of
> source annotations in higher layer computing.
>
> About GNMT in the introduction, we thank you for the scrupulous correction. We
> have deleted it from the paper.
>
> In equation 1, s_i^L is defined in the following text, and the specific meaning
> of variable L is defined in the end of the following text and equation 2.
>
> You concerned Equation 3 forbid a null alignment, which is indeed an advantage
> for FastAlign compared with NMT. This definition is located in the section
> called preliminaries, because previous researches of alignment on NMT is such
> defined. Effective null alignment can definitely improve the alignment
> precision. However, this is not trivial, and there is not effective methods to
> generate null alignment. With our methods, the null alignment can be generated
> according to CFT words. Unfortunately, the AER becomes 44.84 with null
> alignment, which is worse than 41.77 without null alignment. The reason is null
> alignment can not only increase the precision, but also decrease the recall.
> Although null alignment may helpful, it is hard to determine which target word
> should be aligned to null. In addition, comparing AER of CFS words in Table 3,
> FastAlign is still better but with smaller gap than overall words. This means
> except the advantage of null alignment, FastAlign is indeed better.
>
> Thank you for mention the tricks for achieving better alignment. In our
> training, we do not use any of these tricks for both FastAlign and NMT models.
>
> Sorry for confusing you about the description of prediction difference method,
> and we have revised the description in section 3.2. Equation 7 and 8 are
> probabilistic definition of the prediction difference, where an expectation has
> to be approximated by Monte Carlo approach, but equation 9 is a deterministic
> approximation of the expectation. And we have verified the deterministic method
> can achieve the similar performance as the Monte Carlo sampling method with
> enough samples, but only need the same computation as sample size equals one in
> sampling method.
>
> Throughout out this paper, we report AER on NIST 2005 test set, whose reference
> alignment was manually annotated by experts. And we have added this description
> in section 4.1.
>
> You mentioned "translation recall" in the section 4.4. In this section, we
> investigate the performance of alignment and translation based on AER and
> translation recall, which is related to the 1-gram BLEU in some sense. We agree
> your concerning, and we have modified the last sentence in section 4.4 a weaker
> tone.
>
> About the typo of "with replacement" in section 4.5, we have revised it in the
> paper. Thank you very much!

---

> > ### Comment · AnonReviewer1 · 2018-11-15
> > **Thanks**
> >
> > Thanks for your updated paper, and your response to my review. The new 3.2, supported by the experiments in Table 3, greatly helps to clarify the connection between your method and Zintgraf et al.’s method.

---

### Official Review · AnonReviewer3 · 2018-11-02
**Need more convincing**

**Rating:** 5
**Confidence:** 4

**Review:**

The authors study various NMT models and show that most of them do not learn a good alignment model but still achieve good translation. They suggest that the translated output contains two types of words (i) contributed from source (CFS) and (ii) contributed from target (CFT) and hypothesize that even if the alignment is not good the CFT words contribute to the the better quality of translation (as these are not dependent on alignment).

I have a few questions for the authors:

1) Section 3 is titled "Proposed Methods for Inducing Word Alignment". This gives an impression that you are proposing some methods. However, the methods listed in  section 3.2 is from existing work (as you have already acknowledged by citing them correctly). I request you to rename the section or make this more explicit.

2) I am not very convinced about the idea of using a 0 vector to dropout a particular word. Did you try any other options for this ?

3) R_0 and R depend on the model. I mean these scores are as good as the model itself. Therefore I am not sure if it is correct to use these scores to determine CFT and CFS words. Ideally, CFT and CFS words should be a global phenomenon irrespective of the model being used. For example, while translating from a language which does not use articles (a, an, the) to a language which uses such articles, all articles in the target language would be CFT words, irrespective of the model being used. However, this is not the case in your definitions (the definitions are tied to the model scores).

4) Shouldn't you also consider a certain margin  in Equation 11. I mean shouldn't the LHS be greater than the RHS by a certain margin for the word to be classified as CFS.

5) Conceptually, I don't agree with the idea that a word is either CFS or CFT. What about words which require both source and target information? Equation 11 does not completely account for it because the quantity on the LHS does depend on the target words and the quantity on the RHS does dependent on the source words. It is difficult to isolate of effect on source/target.

6) Can you comment on the quality of the silver alignments as the analysis presented in the paper will depend on the quality of these alignments.

---

> ### Author Response · Authors · 2018-11-15
> **New manuscript with more experimental results has been uploaded**
>
> Thanks for your the thoughtful review with interesting suggestions. We address
> your concerns below.
>
> (1) About the title of section 3, it is meant to the new methods for inducing
> alignment, although the methods in section 3.2 is inspired by Zintgraf, who use
> similar method to visualize CNN for image classification.  However, different
> from Zintgraf's sampling method, we purpose a deterministic method. We have
> updated the method descriptions in section 3.2 supported by experiments in
> Table 3. And we have also modified the title to "Methods for Inducing Word
> Alignment". Thanks for motivating us to differentiate our method from
> Zintgraf's sampling method.
>
> (2) Using 0 vector is a deterministic method to approximate the expectation in
> equation 8. We also use the Monte Carlo approach to approximate the
> expectation. Specifically, we sample several words from vocabulary, and use
> their embedding instead of 0 vector to approximate the expectation. As shown in
> the following table, the alignment performance is improving as growing of the
> sample size when using Monte Carlo Method. Fortunately, the result of
> deterministic method, namely using a 0 vector, is very close to the result of
> using Monte Carlo approach with enough sample size. In particular, the
> deterministic method is one order of magnitude faster than the sampling method
> with the best configuration in Table 3, while maintaining comparable AER.
>
> Methods           |                    Sampling PD |||||               | Deterministic PD
> --                | --    | --     | --     | --     | --     | --     | --
> Sample size       | 1     | 2      | 4      | 10     | 20     | 50     | N/A
> AER               | 44.92 | 43.30  | 42.42  | 41.95  | 41.83  | 41.73  | 41.77
> Variance          | 0.004 | < 1e−5 | < 1e−5 | < 1e−5 | < 1e−5 | < 1e−5 | N/A
> Speed (Words/Sec) | 114   | 58     | 27     | 12     | 5      | 2      | 115
>
> (3) You thought CFT and CFS words are global and independent on context.
> Mostly, you are right. At the beginning, we also believe CFT words should be
> very similar with functional words. However, it is hard to define the border
> between CFS and CFT words. For example, we can call higher frequent words CFT
> words and lower frequent words CFS words. But this definition has many
> problems. In addition, some words are polysemous such as "being", which can
> either be a part of "human being" and "having being". Obviously, the meaning of
> this word should be determined by the context. The original partition of CFS
> and CFT words are relevant to a specific model, because it is used to interpret
> the model's prediction. When comparing different models, we also purpose a
> partition of CFS and CFT based on PMI, which is independent on a model being
> used.
>
> (4-5) You thought there should be a probability margin when dividing CFS and
> CFT words. We have generalised the definition of CFS and CFT words in equation
> 11 by introducing a margin. The results of different margin as shown in the
> following table. As shown in this table, different margin generate the
> different partitions of CFS and CFT words. As growing of the margin, the
> partition of CFS and CFT words becomes more confident. And the more confident
> CFS words can achieve better alignment performance and translation performance.
> But the translation recall is still similar between CFS and CFT words, despite
> the big gap of AER between CFS and CFT words. Besides, Contributing From Source
> or Target means where the most contribution comes from, but not means the
> contributions only come from source or target.
>
> Margin | Target Words | AER   | Translation Recall | Proportion
> --     | --           | --    | --                 | --
> 1e-4   | CFS          | 31.64 | 65.54              | 65.61%
>        | CFT          | 62.91 | 66.39              | 24.66%
> 1e-3   | CFS          | 30.33 | 67.82              | 60.40%
>        | CFT          | 63.29 | 69.40              | 22.04%
> 1e-2   | CFS          | 28.26 | 71.56              | 51.53%
>        | CFT          | 64.22 | 73.76              | 17.56%
> 1e-1   | CFS          | 22.87 | 78.59              | 34.85%
>        | CFT          | 64.13 | 78.33              | 10.39%
>
> (6) You mentioned the relationship between the quality of the sliver alignments
> and the analysis in this paper. Firstly, the only analysis relying on sliver
> alignments are the results of explicit model. The better silver alignments will
> only lead the explicit model perform better. And we would like to convey the
> explicit model has achieved enough alignment performance to indicate the
> translation model processes the information of a good alignment.

---

### Official Review · AnonReviewer2 · 2018-11-02
**Sorry I don't fully appreciate the motivation**

**Rating:** 4
**Confidence:** 4

**Review:**

This paper empirically evaluates whether NMT can predict word alignment. This is done by measuring the alignment error rate between silver-data generated from FastAlign and various methods to extract alignment from NMT. The conclusions are that NMT attention does not predict alignment well, and the proposed method of training an additional alignment extraction model performs better.

Unfortunately, I do not appreciate the motivation of the work. Attention is not alignment. Yes, we can try to extract alignment from attention or other parts of the model. But we should really not expect attention to do alignment, because it was simply not designed to do so.

So I am not surprised by the first result that AER for attention-based alignment is bad. I am also not suprised by the second result where training an explicit word alignment model (Eq 6) gets better AER, because that is what the model is designed for.

I do understand that probing alignments might increase our ability to understand and interpret these NMT models. I also thought the CFS and CFT analysis was interesting. But unfortunately I don't think the overall problem attacked by this paper is sufficiently well-motivated for a full paper at ICLR. I do think this will be suitable at, for example, a workshop on visualizing/interpreting neural networks in applications.

Additional note: Please be more specific whether the AER results in Sec 4.2-4.4 are based on scoring on gold alignments in the NIST2015 data, or silver alignments from FastAlign. The former is fine, but the latter might make the results biased for the explicit alignment model trained on the same alignments.

---

> ### Author Response · Authors · 2018-11-15
> **Response to the surprising point of this paper**
>
> Thanks for your feedback. We address your concerns below.
>
> You mentioned you are not surprised by the bad attention-based alignment. Yes,
> you are right, and several previous papers [1,2,3] also addressed
> attention-based alignment is not good enough. However, as far as we know,
> inducing alignment from attention has being the most mainly method to
> investigate the alignment performance of a NMT model in previous researches,
> and several previous works [4,5,6] tried to improve attention-based alignment
> to improve the translation quality. In addition, we were trying to emphasize
> that attention-base alignment may be not only bad but also surprisingly
> terrible. In Table 1, alignment inducing from 6-layers Transformer is even
> worse compared with alignment inducing from PMI. As PMI measures the intrinsic
> alignment from the bilingual training data, this indicates attention-based
> alignment sometimes does not even achieve the amount of alignment information
> from the bilingual data. Therefore, we claim that the inducing alignment from
> attention cannot be regarded as a proper alignment model in general. However,
> NMT indeed can learn alignment from translation as it can be induced by
> prediction difference in spite of not by attention.
>
> You also mentioned you are not surprised by the better alignment inducing from
> explicit model, because you think the model is trained for achieving better
> AER. Yes, you are definitely right, but this is only one of the reasons why
> explicit model achieve better AER. Actually, we investigated the AER of
> explicit model over different pre-trained translation models. We found a bad
> translation model generally also hard to generate good alignment with only
> optimizing the bridge matrix in explicit model. For example, a six layer
> transformer can achieve around 47 BLEU points, and a single layer transformer
> can only achieve around 37 BLEU points. When we train the bridge of explicit
> model over each initialization, the six layer transformer's AER is around 39,
> but the single layer transformer's AER is only around 54. This means training
> an explicit model over a translation model is to unearth the potential of
> inducing a good alignment from the translation model. In other words, if a
> translation model does not possess enough information of a good alignment, it
> is also hard to get a good AER by explicit model.
>
> Finally, thank you for your concerning of the reference alignment. Throughout
> this paper, we evaluate AER by golden alignment labeled by experts.
>
> Reference
> 1. Modeling coverage for neural machine translation
> 2. What does Attention in Neural Machine Translation Pay Attention to?
> 3. Six challenges for neural machine translation
> 4. Guided alignment training for topic-aware neural machine translation
> 5. Supervised attentions for neural machine translation
> 6. Neural machine translation with supervised attention

---

> > ### Comment · AnonReviewer2 · 2018-11-15
> > **explicit model on different transformer/rnns**
> >
> > Regarding your second paragraph in the response: "Actually, we investigated the AER of explicit model over different pre-trained translation models." I think this would be an interesting result but is it in the paper? I couldn't find it. I may have missed it, though. My understanding is that both the explicit alignment and difference prediction method was only applied to Transformer-L6 in the paper (e.g. Table 2 and 3). To really show this point, it would be interesting to compare more comprehensively a variety of MT models with different BLEU/AER characteristics, and then apply the explicit alignment to see how those numbers change.

---

> > > ### Author Response · Authors · 2018-11-16
> > > **New manuscript with more experimental results has been uploaded**
> > >
> > > Thank you for the constructive suggestion.
> > >
> > > We have added the experiments to compare the alignment performance on different
> > > translation models in section 4.3. Because training multiple-layers RNN is too
> > > slow, we investigate Transformer with layers ranging from 1 to 6 in this
> > > experiment. As shown in the following Table, with the same amount of
> > > supervision from silver alignment data, Transformer-L1 generates much worse
> > > alignment than Transformer-L6 and it is only comparable to PMI (fast-align) in
> > > AER. This suggests that supervision is not enough to obtain good alignment and
> > > the hidden units learned by a translation model indeed implicitly capture
> > > alignment knowledge.
> > >
> > >
> > > Transformer | L1    | L2    | L3    | L4    | L5    | L6    | PMI (FastAlign)
> > > --          | --    | --    | --    | --    | --    | --    | --
> > > AER         | 54.50 | 47.94 | 40.47 | 38.40 | 38.80 | 38.88 | 60.18
> > > BLEU        | 36.51 | 44.83 | 45.63 | 47.19 | 46.35 | 46.95 | N/A

---

> > > > ### Comment · AnonReviewer2 · 2018-11-20
> > > > **thanks**
> > > >
> > > > Thanks for adding this. This result does make your argument stronger. I would also recommend showing the AER of the Transformer attention itself in the same table for comparison.
> > > >
> > > > (Note: The results for PMI 60.18 and N/A is flipped in this table.)

---

> > > > > ### Author Response · Authors · 2018-11-21
> > > > > **New manuscript with updated table3 has been uploaded**
> > > > >
> > > > > Thanks for your comment! We have added AER of the Transformer attention to the table3 in the new manuscript.

---

### Comment · Area_Chair1 · 2018-11-19
**Reviewers: Please consider author response**

Hello Reviewers! The authors have made a response that clarify some concerns with more experimental results. Would you be able to take another look and see if this alleviates concerns?

---

### Meta-Review · Area_Chair1 · 2018-12-13
**Important topic, but clarity and framing are somewhat lacking.**

**Confidence:** 4
**Recommendation:** Reject

**Metareview:**

This paper examines the relationship between attention and alignment in NMT. The reviewers all agreed that this is a valuable topic that is worth thinking about.

However, there were concerns both about the clarity of the paper and the framing with respect to previous work. First, it was hard for some reviewers to understand exactly what the paper was trying to do due to issues of the paper structure, etc. Second, there are a number of previous works that also examine similar concepts, and the description of how the proposed method differs seemed lacking.

Due to these issues, I cannot recommend it for acceptance in its current form.